# De novo sensorimotor learning through reuse of movement components

George Gabriel[1]*, Faisal Mushtaq[1,2,3]*, J. Ryan Morehead[1,4]

1 School of Psychology, University of Leeds, Leeds, United Kingdom, 2 NIHR Leeds Biomedical Research Centre, Leeds, United Kingdom, 3 Centre for Immersive Technologies, University of Leeds, Leeds, United Kingdom, 4 Boston Fusion Corporation, Lexington, Massachusetts, United States of America

* g.a.gabriel@leeds.ac.uk (GG); f.mushtaq@leeds.ac.uk (FM)

**Data Availability Statement:** All data and analysis scripts used in this paper are available at https://doi.org/10.17605/OSF.IO/VPFZ5. A video of an

## Abstract

From tying one's shoelaces to driving a car, complex skills involving the coordination of multiple muscles are common in everyday life; yet relatively little is known about how these skills are learned. Recent studies have shown that new sensorimotor skills involving re-mapping familiar body movements to unfamiliar outputs cannot be learned by adjusting pre-existing controllers, and that new task-specific controllers must instead be learned "de novo". To date, however, few studies have investigated de novo learning in scenarios requiring continuous and coordinated control of relatively unpractised body movements. In this study, we used a myoelectric interface to investigate how a novel controller is learned when the task involves an unpractised combination of relatively untrained continuous muscle contractions. Over five sessions on five consecutive days, participants learned to trace a series of trajectories using a computer cursor controlled by the activation of two muscles. The timing of the generated cursor trajectory and its shape relative to the target improved for conditions trained with post-trial visual feedback. Improvements in timing transferred to all untrained conditions, but improvements in shape transferred less robustly to untrained conditions requiring the trained order of muscle activation. All muscle outputs in the final session could already be generated during the first session, suggesting that participants learned the new task by improving the selection of existing motor commands. These results suggest that the novel controllers acquired during de novo learning can, in some circumstances, be constructed from components of existing controllers.

## Author summary

Real-world skills often involve continuous coordination of multiple muscles. Recent studies of sensorimotor skill learning have argued that these skills are not learned by adjusting existing control policies, instead requiring a new controller to be learned "de novo". It remains unclear how such controllers are learned for tasks involving unfamiliar combinations of body movements. In this study, we used a novel human-computer interface task to test this. Over five sessions, participants learned to trace a series of cursor trajectories by coordinating the activation of two muscles. We found that participants tended to reuse

example trial with post-trial feedback is available from https://doi.org/10.17605/OSF.IO/M76G4.

**Funding:** GG was supported by an ESRC White Rose Doctoral Training Partnership Advanced Quantitative Methods Scholarship (https://wrdtp.ac.uk/). GG and FM are supported in part by the National Institute for Health and Care Research (NIHR) Leeds Biomedical Research Centre (NIHR203331; https://leedsbrc.nihr.ac.uk/). The views expressed are those of the authors and not necessarily those of the NHS, the NIHR or the Department of Health and Social Care. The funders had no role in study design, data collection and analysis, decision to publish, or preparation of the manuscript.

**Competing interests:** The authors have declared that no competing interests exist.

the same muscle contractions for trained and untrained variants of the task, and that performance improvements were attributable to improvements in the choice of muscle contractions from a pre-existing repertoire. Our results suggest that new control policies can sometimes be constructed from elements of existing ones.

## Introduction

Sensorimotor control tasks often involve the coordination of multiple muscles [1,2]. From tying shoelaces to driving a car, precise and reliable activation of non-synergistic muscles is a common requirement of everyday tasks. An extensive literature on sensorimotor adaptation explains how well-practised movements can be adjusted to counteract a perturbing influence [3]; but recent studies of human sensorimotor skill learning show that novel coordinated control tasks cannot be learned through adaptation alone [4–7]. Instead, a new controller must be learned in a process termed "de novo" learning (literally, "learning anew"). Despite the importance of de novo learning for the development of everyday sensorimotor skills, relatively little is known about how new controllers are learned.

Existing studies of de novo learning suggest at least three ways in which a new controller may be learned. Firstly, the participant may learn to generate entirely new motor commands. When the repertoire of commands that the participant can already generate does not contain any that are suitable, the participant must learn to generate new commands. Pre-existing neural constraints may prevent or slow the learning of new commands [8,9], and extended practice may be required even when these constraints are surmountable [10]. Secondly, the participant may learn that a given task goal can be achieved using an existing motor command [11]. When suitable commands already exist in the participant's repertoire, but the association between the commands and the resulting output behaviour is unknown, improvements in task performance may be facilitated by trial-and-error exploration of the repertoire [12]. Thirdly, the participant may improve the speed and reliability with which task-appropriate commands are selected from their existing repertoire. When suitable commands already exist in the participants' repertoire, and their suitability for the current goal is known, learning a new controller may still involve improvements to the speed and accuracy with which those commands are produced.

Typical studies of de novo learning in humans attempt to distinguish the influence of some of these learning mechanisms using arbitrary re-mappings of well-practiced body movements to task feedback [5,13,14]. In these tasks, participants typically control the position of a computer cursor using a non-intuitive mapping from body state to cursor position. Studies of this type can reduce or remove the component of learning new motor commands by designing the mapping so that existing motor commands are sufficient to support the execution of the task. While studies of this sort have demonstrated that the relatively long timecourse of de novo learning (compared to adaptation) is not exclusively attributable to learning new motor commands, the generality of these findings may be limited by the design of the studies.

One common limitation of de novo learning studies relates to the temporal component of sensorimotor skill. In many real-world tasks, appropriate relative timing of activity across multiple non-synergistic muscles is essential for effective execution of the task. In contrast, for de novo learning tasks with discrete task goals, relative timing of individual muscle outputs may have little bearing on whether the goal is achieved. For example, some tasks which re-map multiple limb positions to lower-dimensional cursor position can be executed by sequentially moving each limb, with the requirement for simultaneous coordination of the movements

enforced only implicitly by time constraints. Using a continuous control task which directly requires co-activation of non-synergistic muscles may overcome this limitation and help to explain how new controllers emerge in a more general class of learning scenarios.

An additional limitation of existing de novo learning studies arises specifically from their use of well-practiced movements. When forming a new association between an existing movement and a task goal, learning may be slowed by interference from prior associations [14]. As the new task's target associations are deliberately perturbed relative to pre-learned ones, the tendency to reuse pre-learned associations can hinder learning. An ideal de novo learning paradigm should distinguish the influence of interference on learning rate from the influence of the intrinsic processes involved in building a new controller.

To contribute to addressing these limitations, we developed a novel de novo learning paradigm in which participants learned an unfamiliar continuous control task requiring precise temporal and spatial coordination of non-synergistic muscles. Participants controlled the horizontal velocity of a computer cursor using two EMG signals: one from a muscle of the right hand and one from a muscle of the left shin. Muscle activity was mapped to cursor velocity via a redundant mapping, allowing individual participants to develop idiosyncratic controllers. Half the participants were assigned a congruent mapping, in which the laterality of the muscle on the body matched the direction of that muscle's contribution to cursor velocity. The remaining half used an incongruent mapping in which the mapping directions were reversed, but the muscle laterality was the same. Over five sessions on five consecutive days, participants practiced following two cursor trajectories with post-trial visual feedback. Participants also practiced a further four trajectories without visual feedback, three of which required reversed order of muscle activation relative to the trained conditions. Over the five sessions, we observed improvements in both the shape of the trained cursor trajectories and the timing of their peaks relative to the target trajectory. Peak timing, but not trajectory shape, also showed consistent improvements in all untrained conditions. Despite the observed improvements in performance, the per-channel outputs generated in the final session by each participant could already be generated during the first session. Qualitatively similar patterns of improvement were observed for participants in both the congruent and incongruent groups, though learning in the latter group was slower. These results are consistent with learning to reliably select appropriate motor commands from a pre-existing repertoire.

## Materials and methods

### Ethics statement

The University of Leeds School of Psychology ethics committee granted ethical approval for this study. Written informed consent was obtained for all participants via the web-based sign-up form.

### Participants

A total of 20 participants (age 19–35, median 23 years; 12 male) each completed one session per day of an electromyographic control task for five consecutive days. Participants completed a pre-session questionnaire describing their prior experience with playing computer games, playing musical instruments, participating in sports, and driving. This information was not used to select participants for inclusion or exclusion from the study. All participants had no known neurological disorder and provided written consent through the online study sign-up process.

Participants were assigned to one of two conditions, labelled "congruent" (10 participants, 6 male) or "incongruent" (10 participants, 6 male). The two groups completed identical

sessions but each used a differently configured myoelectric interface. During post-hoc analysis, one male participant was excluded from each group due to having extremely large mean cursor amplitude (more than double the maximum required amplitude) or mean cursor peak time after the end of each trial. The presented analyses use the remaining 18 participants unless otherwise stated.

Participants completed the Edinburgh handedness inventory, and we allowed participation regardless of handedness. Two participants (one male, one female) from the incongruent group and one female participant from the congruent group were identified as strongly left-handed.

## Experiment setup

Participants sat on a chair with both feet placed on blocks and arms resting on armrests (Fig 1). Abduction of the fingers of the right hand and dorsiflexion of the left foot were limited using Velcro straps. The straps were placed around the fingers of the right hand and over the bridge of the left foot. Participants self-adjusted each strap to the maximum comfortable tension at the start of each session. Participants viewed visual feedback for the experiment tasks on a computer monitor (Dell AW2521HFLA, 24.5-inch, 1080 × 1920 pixels, 244Hz) at a distance of approximately 1.1 metres. The framerate of the task feedback was consistently above 110Hz.

Two bipolar EMG channels were recorded in real-time at 2048Hz using a biosignal amplifier (OTBioelettronica Quattrocento) and custom interface code written in Python. One channel was recorded from the left shin (tibialis anterior) and one from the right hand (abductor digiti minimi) of each participant. The shin electrodes were centred at points 7cm apart along a vertical line approximately 6cm below the tibial tuberosity and 2cm to the lateral side of the anterior margin of the tibia. The hand electrodes were centred at points 3cm apart approximately equidistant from the pisiform bone and the base of the fifth metacarpal bone. A reference electrode was placed on the ulnar styloid process of the right wrist. Locations of the electrodes were marked on the skin in ink and re-marked each session to allow consistent placement of the electrodes. All electrodes were self-adhesive solid gel type (Skintact F-261, 26mm diameter), and were further secured using micropore tape (hand electrodes) or kinesiology tape (shin electrodes).

## Experiment tasks

Each session comprised a series of calibration tasks and experiment tasks. Detailed instructions were presented to the participants through simultaneous on-screen text and audio narration. Instruction transcripts are available in the experiment data repository. All sessions were identical in structure except for the addition of a single practice block and more detailed instructions in the first session.

**Calibration.**   To set the power range of the two muscles, participants completed a maximum and minimum contraction task at the start of each session. The values recorded during the first session were used to set the gain of the electromyographic interface for all sessions. Maximum and minimum power level data from other sessions was used to track cross-session changes in signal-dependent noise for post-hoc analysis, but the gain of the interface was not changed after the first session.

Before calibration, participants were shown videos demonstrating how to activate the target muscles through abduction of the right little finger or dorsiflexion of the left foot. Calibration was completed separately for each of the two muscles. During calibration of a muscle, participants were shown a streaming lineplot on screen representing the instantaneous smoothed

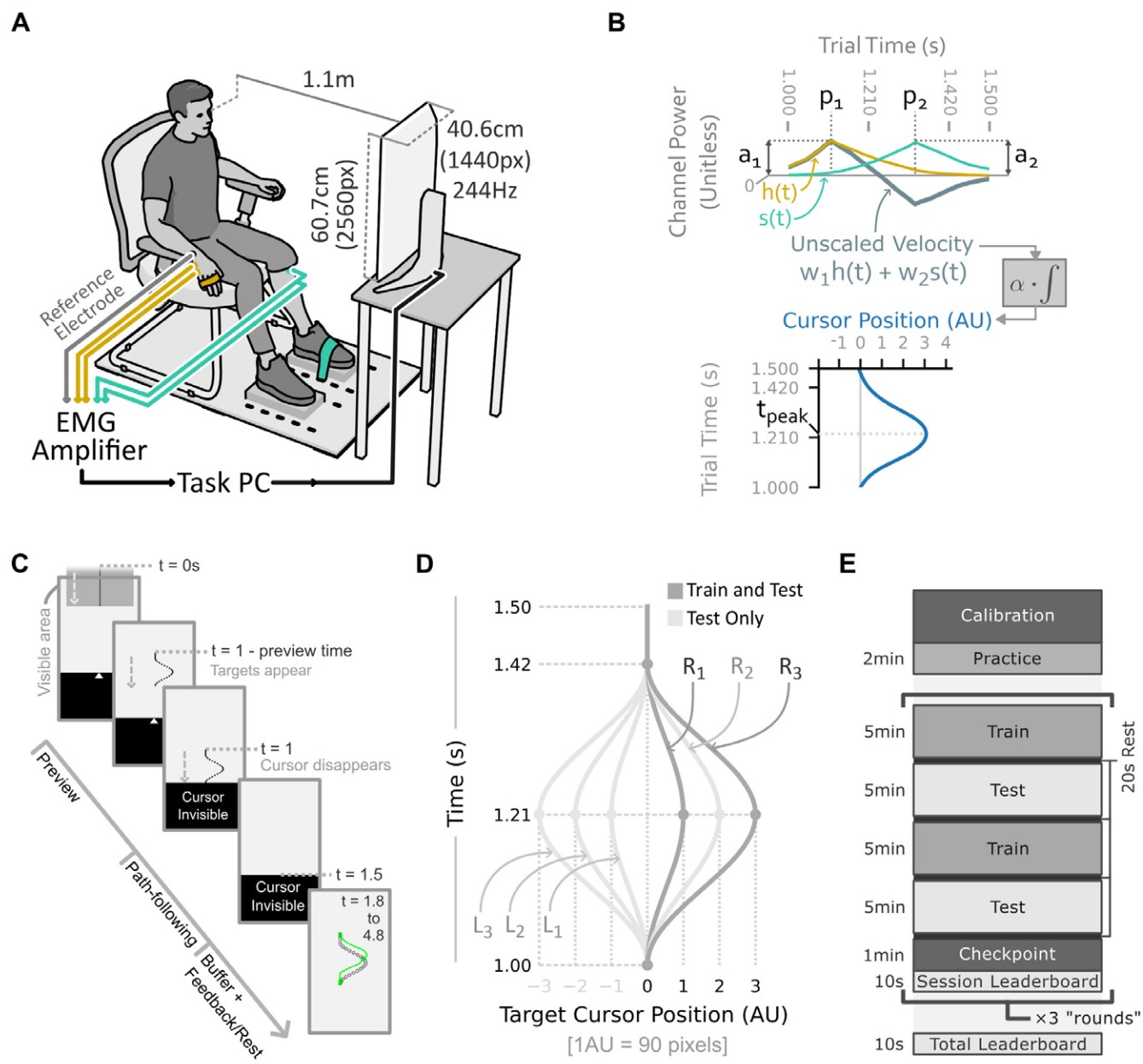

**Fig 1. Experiment setup, EMG interface, and task design** (A) Participants sat in a chair 1.1m from the computer screen on which task feedback was displayed. A bipolar EMG channel was recorded in real time from the right abductor digiti minimi (yellow) and the left tibialis anterior (green), and used to control the horizontal velocity of a cursor. (B) The smoothed and scaled EMG signals generated by the hand, $h(t)$, and shin, $s(t)$, were combined in a weighted sum to produce the unscaled cursor velocity. The output cursor trajectory relative to centre was given by the integral of this velocity signal multiplied by a constant velocity scale factor, $\alpha = 2500$ pixels·s$^{-1}$. (C) Each trial of the main task proceeded through several stages, as described in the main text. In training trials, feedback of targets hit (green circles) and produced cursor trajectory (green curve) were given for 3s. On probe trials, no feedback was given, and participants rested for 3s. (D) Six different path shapes were used in the main task, named according to their direction (Right or Left) and magnitude (1, 2, or 3). (E) The schedule of trial blocks used in each session of the study. On session 1 an initial instructions block was also included at the beginning.

EMG power from that muscle. On maximum contraction trials, participants were instructed to contract the target muscle as strongly as possible using the demonstrated action, and to hold the contraction until told to release. The contraction stage lasted ten seconds from commencement of instructions to completion, and the mean of the smoothed EMG signal over the final

four seconds of the contraction was set as the maximum power level of the muscle. The maximum contraction task could be repeated at the discretion of the experimenter if the participant appeared to have performed a sub-maximal or inconsistent contraction upon inspection of the EMG trace.

In the minimum power calibration task, participants were instructed to relax the muscle as fully as possible. This stage lasted fifteen seconds, and the mean of the smoothed EMG signal over the final ten seconds of data was used to define a noise threshold for the muscle. This task could also be repeated at the discretion of the experimenter if movement or incomplete relaxation of the muscle was suspected.

Checkpoints were used after every four blocks of trials to re-assess the baseline noise level using the above-described minimum power calibration method. If the noise level during a checkpoint was found to be greater than the original baseline, the experimenter checked electrode adhesion and participant seating position and repeated the checkpoint.

**Power cycle.** To allow participants to calibrate the strength of their muscle contractions to the range required in the main tasks, a "power cycle" task was completed after calibration in each session. For this task, one of the two EMG channels was selected, and the scaled power in that channel was used to control the vertical position of a small black circle. A larger grey circle moved with sinusoidal velocity up and down a line of 640 pixels height for three cycles over 50 seconds. The participant was instructed to move the black circle to keep it inside the grey circle by flexing the target muscle. Online visual feedback of the position of both circles was provided throughout. The same task was completed for each muscle in each session.

**Free movement and path following tasks.** The main task in each session involved controlling the velocity (not position, as was the case in the power cycle task) of an on-screen cursor. The cursor was constrained to move horizontally in the "cursor zone" vertically positioned at 1/3 screen height. To allow participants to familiarise themselves with the interface before starting the main path following task, a 30 second "free movement" stage was included after calibration in each session. During this stage, participants were shown the cursor and allowed to practice moving it using the velocity control interface. Online visual feedback of the cursor position was provided throughout.

In the main task (Fig 1C), participants were shown a grey box descending towards the cursor zone at a constant speed. The box descended for one second before reaching the cursor zone, after which it continued descending through the cursor zone for 0.5s. At either 0.5s or 0.8s before arrival at the cursor zone, the box was replaced with a curving target path of equal height. This period is referred to as "preview time". The path was represented by a series of 25 circular targets of six-pixel radius, uniformly vertically distanced along the shape of the path. Participants were instructed to move the cursor so that its tip (the highest point of the triangular cursor) touched as many of the descending circles as possible. Participants were specifically instructed to try and hit all targets rather than ignoring some of them. On all trials, the cursor became invisible 0.25s before the path arrived at the cursor zone. Online feedback of cursor position was therefore not available during the path-following segment of each trial, and participants had to hit the targets without seeing the current cursor position. A video of an example trial with post-trial feedback is available from the data repository for this paper (https://doi.org/10.17605/OSF.IO/M76G4). The horizontal cursor position was reset to the centre of the screen at the start of each trial. If the cursor moved more than 10 pixels from the centre of the screen while visible, the trial was abandoned, a warning buzzer sounded, and a written instruction was displayed informing the participant that they moved the cursor too early. Abandoned trials were repeated at the end of the block.

Each trial was followed by either a feedback stage or a no-feedback rest stage of three seconds duration. Performance feedback, when given, consisted of a trace of the actual cursor

trajectory that the participant followed, aligned with the target circles. Hit targets were indicated as green filled circles, while missed targets were indicated as black unfilled circles.

Six different path variants were used, all of which were scaled versions of the same shape (Fig 1D). Three magnitudes of paths corresponding to 90, 180, and 270 pixels amplitude (1, 2, and 3 arbitrary units) and two directions (leftward peak and rightward peak) were used.

Two types of trial blocks were used: training blocks, and test blocks. In training blocks, only the small-rightward and large-rightward paths were used, all trials had 0.8s preview time, and feedback was given after each trial. Training blocks included 30 trials of each of the two conditions, in pseudorandom order. In test blocks, all six path variants were used, both 0.5s and 0.8s preview times were used, and no feedback was given after each trial. Test blocks featured five trials of each of the 12 conditions, pseudorandomised such that the same condition was not repeated with fewer than 3 trials of other conditions between the repetitions. The first session included an additional practice block before the first training block, in which participants practiced one trial of each of the twelve conditions with post-trial feedback.

A single round of the study consisted of two alternating train-test block pairs, interspersed with 20s rest periods, followed by a noise "checkpoint" as described above. Participants were also shown a session leaderboard for 10 seconds at the end of each round, featuring their and other anonymised participants' cumulative numbers of targets hit up to that stage of the corresponding session. Three rounds were completed in each session. At the end of each session, the participants were shown a leaderboard featuring the total number of targets they had hit across all sessions, together with other participants' totals.

Our decision to not provide online visual feedback of cursor position in the path-following task was designed to limit learning during no-feedback trials. These trials were designed to probe generalisation of learning from the two trained path conditions to the untrained conditions. If online visual feedback had been provided during both the training and test blocks, changes in performance on the untrained conditions could have resulted from the visual feedback received during no-feedback trials. To ensure that the tasks used in the training and test blocks were consistent, we therefore omitted online visual feedback in both cases, using post-trial feedback for only the trained conditions.

## EMG interface

To create a low-latency control signal using the bipolar EMG signals, on each frame, a weighted average of the latest samples of rectified EMG data was computed for each channel. Two variants of the interface were used for different tasks. For the free movement and path following tasks, a weighted average of the rectified bipolar EMG was taken using a 256-sample triangular smoothing kernel which assigned greatest weight to the most recent EMG sample. For the power cycle task, a longer uniformly weighted kernel of 4096 samples was used. In both cases, the noise threshold for each channel (as identified during calibration) was subtracted from its smoothed EMG signal, and resulting values less than zero were set to zero. The thresholded and smoothed EMG channels were then scaled such that 35% of the participant's maximum power level produced an output signal of 1. We refer to the resulting time-varying signals as the channel profiles, $h(t)$ and $s(t)$, from the hand and shin muscle respectively (Fig 1B).

Two variants of the thresholding method were used for different tasks. For the main free movement and path-following tasks, a log-normal distribution was fitted to the baseline EMG data recorded during calibration, and the noise threshold was set at the 99.99th percentile of this distribution plus 1% of the EMG power at maximal contraction. This thresholding method provided robustness to noise without excessively reducing the dynamic range of the control signals. For the power cycle task, only the 1% of maximal contraction

threshold was used. This was chosen to prevent the introduction of a noticeable "dead zone" in the controller, given that position control rather than velocity control was used in the power-cycle task.

For the path-following task, each control signal was linearly mapped to a one-dimensional velocity value, such that a control signal of 1 resulted in a cursor velocity of 2500 pixels per second. Unscaled control signals with magnitude greater than 1 were not capped. The entire screen width could therefore be traversed in 0.432 seconds without exceeding 35% of maximal power. The two channels each controlled velocity in opposite directions: one for positive (rightward) velocity, one for negative (leftward) velocity. A weighted sum of the two channels' control signals then determined the unscaled velocity of the cursor. The integral of this unscaled velocity signal (multiplied by 2500 pixels per second) gives the cursor trajectory used to complete the path-following task.

For participants from the congruent group, the left shin was mapped to leftward cursor velocity (i.e. negative velocity values), while the right hand was mapped to positive velocity ($w_1$ = 1 and $w_2$ = −1 in Fig 1B). For participants in the incongruent group, the signs of the velocity for each channel were flipped ($w_1$ = −1 and $w_2$ = 1 in Fig 1B). The same laterality of electrode placement (left shin and right hand) was used for both groups.

## Data analysis

**Performance measures.**   To quantify improvements in the cursor trajectory shape independently of its timing relative to the target trajectory, we computed a peak-aligned version of the output trajectories (illustrated in S1 Fig). For each input channel, we defined the peak amplitude of the channel profile as its maximum value occurring between 0.7s and 1.8s after trial start and during any period of more than 16 samples of consecutively non-zero activity (if such a period exists for the given trial). We also defined the channel initiation as the time at which that consecutive interval of samples started. We then defined the peak amplitude of the cursor trajectory as its maximal amplitude occurring between the identified peak times of the two input channels. We next generated an interpolated version of the channel, shifting it in time such that the identified peak occurred at 1.21s after trial start. This interpolation also reduced the sampling rate of the cursor trajectory from 2048Hz to 1000Hz to reduce the computational load for subsequent analyses.

Two basic performance measures were computed using the peak-aligned cursor trajectory. Firstly, the peak-aligned target hit percentage is the percentage of the target points along the cursor trajectory that the peak-aligned trajectory intersected. Secondly, the root-mean-squared peak-aligned cursor trajectory error (denoted $\epsilon$) is the root-mean-squared error between the observed peak-aligned cursor trajectory and the target trajectory for that trial. The time of the cursor trajectory peak (computed before peak alignment) is also used as a basic performance measure.

**RMS error model.**   Two Bayesian regression models are used repeatedly throughout the analyses to produce estimates for the mean of the performance measures and channel features in each session. In all cases, we sampled the posterior distributions for the models using a NUTS Markov chain Monte Carlo sampler, implemented in Python.

For the RMS error, we used a hierarchical model with a common component shared across participants in the same congruence condition. In all applications of the model, we only used data from no-feedback trials where the hand and shin channel peaks were in the correct order (as determined by the target trajectory direction and the participant's congruence condition). We also centred, log-transformed, and re-scaled the values to have a sample standard deviation

of 1 across all participants combined. The model is then as follows:

$$\mu_c \sim Normal(0, 0.75)$$
$$\mu_{p,s,d} \sim Normal(\mu_c[p], 0.75)$$
$$\sigma_{p,s,d} \sim HalfNormal(1)$$
$$y_i \sim Normal(\mu_{p[i],s[i],d[i]}, \sigma_{p[i],s[i],d[i]})$$

Where $\mu_c$ is a congruence group specific parameter, indexed by participant $p$; $\mu_{p,s,d}$ are the specific contributions to mean RMS for each participant, scale condition, and day (session); $\sigma_{p,s,d}$ is the specific standard deviation for each participant, scale condition, and day; and $y_i$ is the (appropriately transformed) RMS of one observed trial.

Trajectory peak time also uses the same model, with $y_i$ representing the (appropriately transformed) trajectory peak time of one observed trial.

To determine if there are differences in RMS error for different preview time conditions in session 1, we use a similar model where $d$ is replaced by $q$, representing either 0.5s or 0.8s preview time.

**Channel peak feature model.**   For the peak amplitudes and times of each channel, we used a simpler model, transforming the data in the same way as for the RMS error model. This model was applied separately to data for each channel, again rejecting trials in which the order of channel peak activation was incorrect.

$$\mu_{\{p,s,d\}} \sim Normal(0, 1)$$
$$\sigma_{\{p,s,d\}} \sim HalfNormal(1)$$
$$y_i \sim Normal(\mu_{\{p[i],s[i],d[i]\}}, \sigma_{\{p[i],s[i],d[i]\}})$$

(2)

Where $\mu_{p,s,d}$ are the specific contributions to mean RMS for each participant, scale condition, and day (session); $\sigma_{p,s,d}$ is the specific standard deviation for each participant, scale condition, and day; and $y_i$ is the (appropriately transformed) channel peak amplitude (or time) of one observed trial.

**Channel peak time correlation model.**   To estimate the correlation between peak times in the first and second-activating channels, we used a multivariate normal model. Prior to model fitting, we subtracted the first channel activity start time from the peak times of both channels in each trial, and re-centred the resulting dataset to have sample mean of zero.

$$\mu_1, \mu_2 \sim Normal(0, 0.1)$$
$$\sigma_1, \sigma_2 \sim Exponential(0.5)$$
$$R \sim LKJcorr(1)$$
$$S = \begin{pmatrix} \sigma_1 & 0 \\ 0 & \sigma_2 \end{pmatrix} R \begin{pmatrix} \sigma_1 & 0 \\ 0 & \sigma_2 \end{pmatrix}$$
$$\begin{pmatrix} p_1 \\ p_2 \end{pmatrix} \sim MvNormal\left( \begin{pmatrix} \mu_1 \\ \mu_2 \end{pmatrix}, S \right)$$

(3)

Where $\mu_1$ and $\mu_2$ are the prior means for the two-dimensional multivariate normal; $\sigma_1$ and $\sigma_2$ are the prior standard deviations for the covariance matrix; $R$ is an LKJ prior for the unit standard deviation covariance matrix; $S$ is the prior over covariance matrices, with scaled standard deviation; and $p_1$ and $p_2$ are the channel peak times for the first and second-activating channels, with initiation time subtracted and re-centred to have sample mean of 0.

**Bayes factors.** The reported Bayes factors are computed from the posterior distributions of the parameters of interest. Unless otherwise stated in the figure caption, the Bayes factors in favour of a reduction in a mean feature $x$ value from session $a$ to session $b$ are computed using the formula:

$$\frac{\Pr(x_b < x_a|X)\Pr(x_b \geq x_a)}{\Pr(x_b \geq x_a|X)\Pr(x_b < x_a)} \tag{4}$$

Where $X$ is the observed data. All prior and posterior probabilities are estimated by sampling from the respective distributions, and computing the proportion of samples satisfying the relevant inequality.

The $BF_{10}$ Bayes factors reported in most of our paper describe the ratio of evidence in favour an alternative hypothesis over a null hypothesis. For example, a $BF_{10}$ of 3 in favour of a reduction in mean RMS error between sessions 1 and 5 says that the evidence in favour of a reduction (under the assumptions of our Bayesian model described above) is three times stronger than the evidence against a reduction. Occasionally, we find it more informative to report a $BF_{01}$ Bayes factor, which gives the ratio of evidence in favour of the null compared to evidence in favour of the alternative hypothesis.

We used Bayes factors instead of more traditional frequentist statistical methods (e.g. ANOVA) for three main reasons. Firstly, Bayes Factors provide a measure of the strength of evidence both in favour of and against the null hypothesis. Typical frequentist statistical tests do not provide this additional information, which is important for assessing the robustness of our conclusions. Secondly, Bayes Factors do not require assumptions such as heterogeneity of variance across the compared groups or normality of residuals, while frequentist methods such as ANOVA do. In our data, it can be readily observed that variance in several of the measures of interest is not consistent between the compared groups, invalidating the use of typical frequentist analyses. Thirdly, Bayes Factors allow the uncertainty in our estimates of the quantities of interest to be incorporated into the hypothesis test. Conclusions drawn from Bayes Factors can therefore be more robust than those drawn from analysis methods which compare only averaged quantities. For further discussion of these points and additional advantages of Bayesian analyses, we refer the reader to Wagenmakers et al. [15].

## Results

### Performance on trained conditions improved gradually over multiple sessions

Participants controlled the horizontal velocity of a computer cursor using bipolar EMG signals recorded from a muscle of the right hand (abductor digiti minimi) and a muscle of the left shin (tibialis anterior). Each participant was randomly assigned to one of two groups: congruent or incongruent. For the congruent group, each channel affected cursor velocity in the direction matching the laterality of the source muscle on the body (i.e., left shin to leftward velocity, and right hand to rightward velocity). For the incongruent groups, the muscles' laterality was unchanged, but the direction of their velocity contributions was reversed.

In the main task, participants were instructed to move the cursor to hit a series of circular targets that descended at a constant speed down the screen. Online visual feedback of cursor position was not provided, but post-trial feedback of target hits and output cursor trajectory was provided during training blocks.

To determine whether the shape of participants' cursor trajectories improved independent of their timing relative to the target trajectory, we aligned the amplitude peaks of the output and target trajectories and computed performance measures based on these peak-aligned

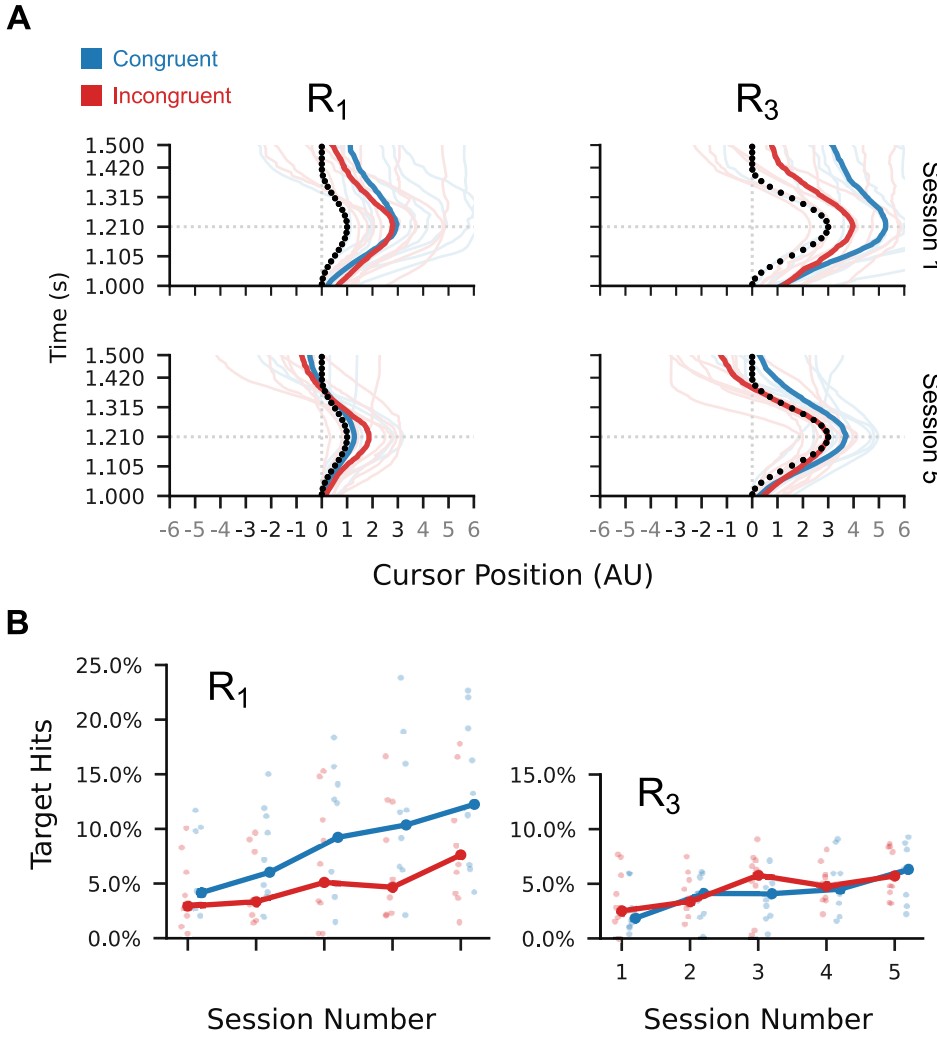

**Fig 2. Example cursor trajectories and peak-aligned target hits.** (A) Session 1 and 5 median cursor trajectories per participant (faint lines) and across participants (dark lines) in the conditions trained with post-trial feedback ($R_1$ and $R_3$). Inset black circles show the target points (to scale) for each trial condition. (B) Peak-aligned target hit percentages for the for the trained trial conditions. Lines show cross-participant medians within each condition group, and points show per-session means for individuals. Session 1 to session 5 increase in median hit percentage was 6.8% (SD = 7.6%) for congruent $R_1$; 4.4% (SD = 5.0%) for incongruent $R_1$; 2.5% (SD = 3.9%) for congruent $R_3$; 2.6% (SD = 3.3%) for incongruent $R_3$.

trajectories. Over five sessions, participants' mean peak-aligned target hit percentages for the trained trajectories ($R_1$ and $R_3$) showed a steady but statistically unreliable increase (Fig 2B). This is likely due to qualitative improvements in cursor trajectory shape which do not consistently result in more targets being hit, as demonstrated by the shape of session 5 median cursor trajectories compared to those of session 1 (Fig 2A, dark curves).

To provide a more sensitive measure of trajectory shape quality, we computed root-mean-squared error between the peak-aligned output and target trajectories. Marginalising across participants, we observed statistically robust reductions in the posterior mean and standard deviation of the peak-aligned RMS error between sessions 1 and 5 (Fig 3, top left plots). These reductions are less robust for the incongruent group, likely because this group had lower

variability in session 1 than did the congruent group: Bayes factors in favour of the incongruent group having lower $\sigma(\epsilon)$ on session 1 than the congruent group are 11.23 for $R_1$, and 5.58 for $R_3$. Bayes factors in favour of the incongruent group having lower $\epsilon$ on session 1 than the congruent group are 1.09 for $R_1$, and 1.47 for $R_3$. Together, these results are consistent with improvements in mean trajectory shape and reductions in the variability of the generated trajectory shape.

A separate feature of performance in the path-following task is the temporal alignment between the generated cursor trajectory and the target trajectory. To quantify how well participants aligned their outputs with the target, we computed the times at which each trajectory reached its largest amplitude (i.e. its peak) in comparison to the ideal peak time (Fig 3, bottom row). The per-participant and cross-participant marginal mean peak times show a statistically robust improvement, as measured by Bayes factors in favour of a reduction from session 1 to session 5. Variability in peak timing may also have reduced between sessions 1 and 5, but this is less statistically reliable for the incongruent group than the congruent group.

## Improvements in peak timing but not trajectory shape transferred to the untrained leftward conditions

To further clarify which learning processes resulted in the observed performance improvements on the trained paths, we assessed how these improvements transferred to the leftward conditions $L_1$ and $L_3$. These conditions were untrained (i.e. practiced without post-trial visual feedback) and required reversed order of input channel activation compared to the rightward conditions. As such, if the learning for the rightward conditions was specific to the trained order of muscle activation, we would expect little transfer to the leftward conditions.

Despite robust reductions in mean peak-aligned RMS for $R_1$ and $R_2$ between sessions 1 and 5, we found relatively weaker evidence in favour of a reduction for $L_1$ and $L_3$ in the congruent group and weak evidence in favour of no change or an increase in the incongruent group (Fig 3, top right). The standard deviation of RMS for the leftward path showed similarly weak evidence of a reduction for both congruence conditions, with Bayes factors around a quarter the magnitude of those observed for $R_1$ and $R_3$.

In contrast, although the leftward path conditions require a different order of input channel activation than the trained rightward conditions, there is robust evidence in favour of an improvement in mean peak timing and a reduction in peak time standard deviation in the leftward conditions for the congruent participants (Fig 3, bottom right). The Bayes factors for the incongruent group also favour an improvement in peak time and a reduction in standard deviation of peak time, with evidence approximately as strong as in the corresponding trained rightward conditions.

## Performance improvements transferred to the untrained medium amplitude rightward condition

The above analyses demonstrate that, despite the unfamiliarity of the task and interface, participants were able to execute the task and, on average, improved their performance on trained conditions over the five sessions. We next sought to determine whether this learning could have arisen due to the formation of habitual responses to the trained trajectories, rather than emergence of a new controller as is purported to occur in de novo sensorimotor skill learning. To achieve this, we assessed whether the performance improvements observed for the trained conditions arose concurrently in the untrained task conditions. We reasoned that, while de novo skill learning could support condition-general improvements in performance, habit formation should not [16].

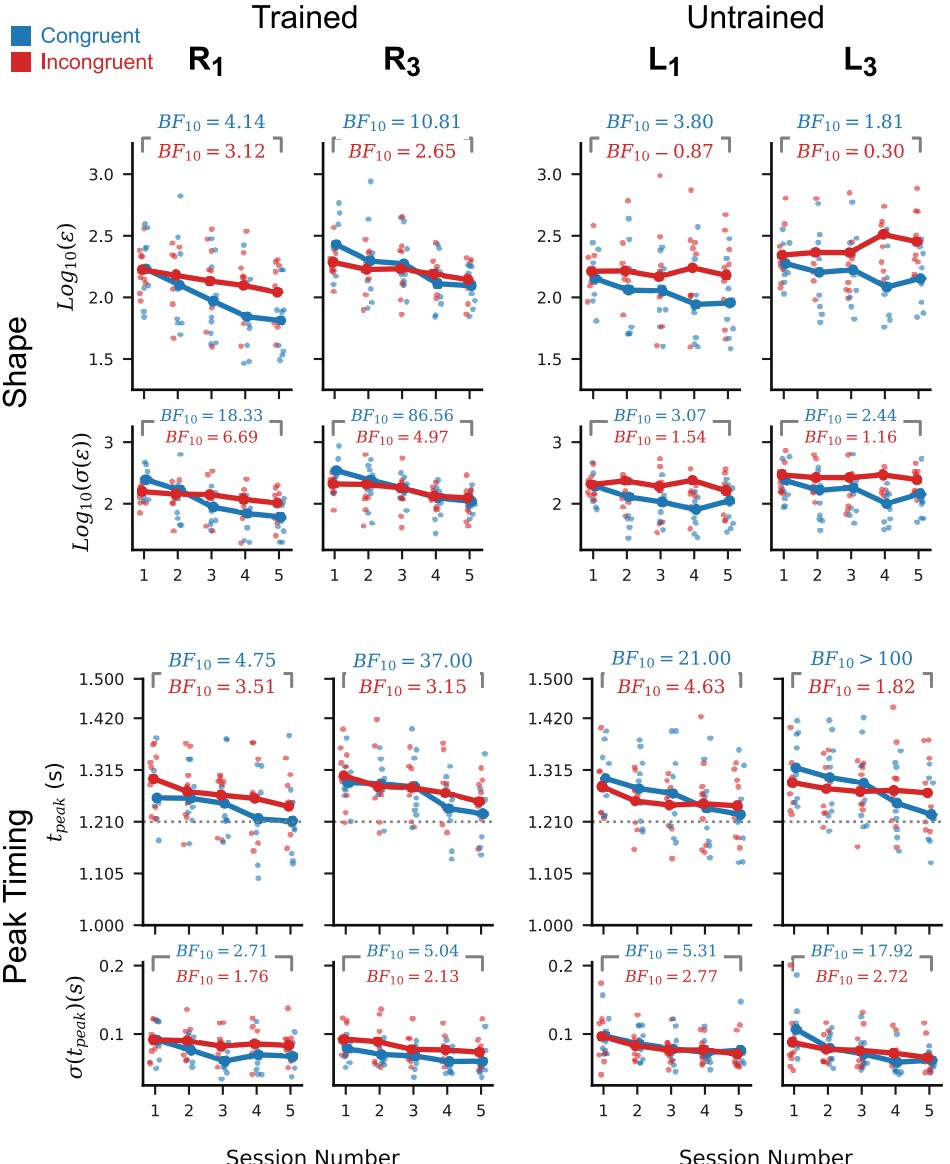

**Fig 3. Task performance on the trained and untrained conditions.** Per-participant posterior means (larger plots, faint points) and standard deviations (smaller plots, faint points) of peak-aligned RMS error (top row) and trajectory peak time (bottom row) in the generated cursor trajectories. Dotted horizontal line in the lower plots represents the ideal trajectory peak time. Marginal means across sessions for the trained conditions ($R_1$ and $R_3$). Only peak timing shows a consistent improvement across participants for the untrained conditions ($L_1$ and $L_3$). Inset numbers are Bayes factors in favour of a reduction in marginal mean statistic from session 1 to session 5 for each of the two participant groups individually. With the exception of the trajectory shape for untrained conditions, Bayes factors for both the congruent (blue) and incongruent groups (red) favour improvements in performance and reductions in variability with respect to both shape and peak timing.

The medium-magnitude rightward path ($R_2$) can be executed by generating EMG outputs with an intermediate magnitude between those used for the small and large rightward paths. Unlike the other three untrained conditions, $R_2$ also requires the same order of channel activation as the trained paths, determined by the participants' congruence conditions. All

performance results for $R_2$ were qualitatively similar to those for $R_1$ and $R_3$: there was a statistically unreliable increase in peak-aligned target hit percentage across the sessions (Fig 4A); the median shape of the output cursor trajectories on "no-feedback" trials improved from session 1 to session 5 (Fig 4B); and we found statistically robust Bayes factors in favour of reductions in the mean and standard deviation of peak-aligned RMS (Fig 4C) and trajectory peak time (Fig 4D) between sessions 1 and 5.

Although performance on $R_2$ improved according to the selected measures, these trends do not show that the outputs being generated for the $R_2$ trials were specific to that condition. Performance improvements for this condition could also arise if participants used the same outputs for $R_2$ as for $R_3$ or $R_1$, as outputs suitable for these conditions will approximate the trajectory shape required by the $R_2$ condition. It is unclear from inspection of the median cursor trajectories for the $R_2$ condition (Fig 4B) whether the generated outputs are distinct from those of the other two conditions. To clarify this, we consider the per-participant distributions of input channel peak amplitudes (not to be confused with cursor trajectory peak amplitudes). These distributions show idiosyncratic differences dependent upon path condition. For some participants there is a clear difference in the amplitude distributions for each of the three conditions in session 5 (S2A Fig), while for other participants the medium rightward trial distribution in session 5 almost perfectly coincides with those of the large or small rightward trial conditions (S2B Fig). This suggests that, while some participants generated condition-specific outputs, others simply re-used one or both outputs from the trained conditions to execute the $R_2$ condition. Grouping participants by whether their peak channel amplitudes had distinguishable or indistinguishable distributions for the three rightward trial conditions, we observed in both cases statistically robust reductions in RMS error (S2C Fig), trajectory peak time (S2D Fig), and the standard deviation thereof. This demonstrates that transfer of performance improvements to the untrained rightward condition was achieved either by production of untrained intermediate outputs or by simple re-use of outputs suited to other conditions.

## Per-channel features were consistently similar for rightward and leftward paths of equal magnitude

The patterns of transfer described in the preceding section suggest that the learning processes responsible for determining trajectory shape and peak timing are partly independent. To clarify how these processes give rise to performance improvements, we now assess learning-related changes in the properties of the per-channel control signals.

Direct comparison of the per-participant mean channel profiles for leftward and rightward trial conditions suggests that the choice of profile shapes played an important role in both effects. In particular, the mean profiles for a given channel tend to be very similar regardless of whether that channel is activated in the context of a leftward or rightward path trial (Fig 5A, light versus dark traces; Data for all participants is shown in S7 Fig). This re-use of output profiles across directions has different consequences for the timing and the shape of the resulting cursor trajectory.

As the hand and shin channel profiles tend to differ in shape (including with respect to their amplitude), the cursor velocity resulting from taking a weighted sum of the two will differ depending on the order in which the profiles are generated. Consequently, the generated cursor trajectory will differ in shape when the same channel profiles are used in reversed order (Fig 5C). Notably, the timing of the cursor trajectory peak is not as strongly affected by reversing the order of the channels. If each channel profile is approximately symmetrical about its peak and has approximately equal duration, activating the two channels at the same times but in reversed order will result in a cursor trajectory that reaches its peak at approximately the same time.

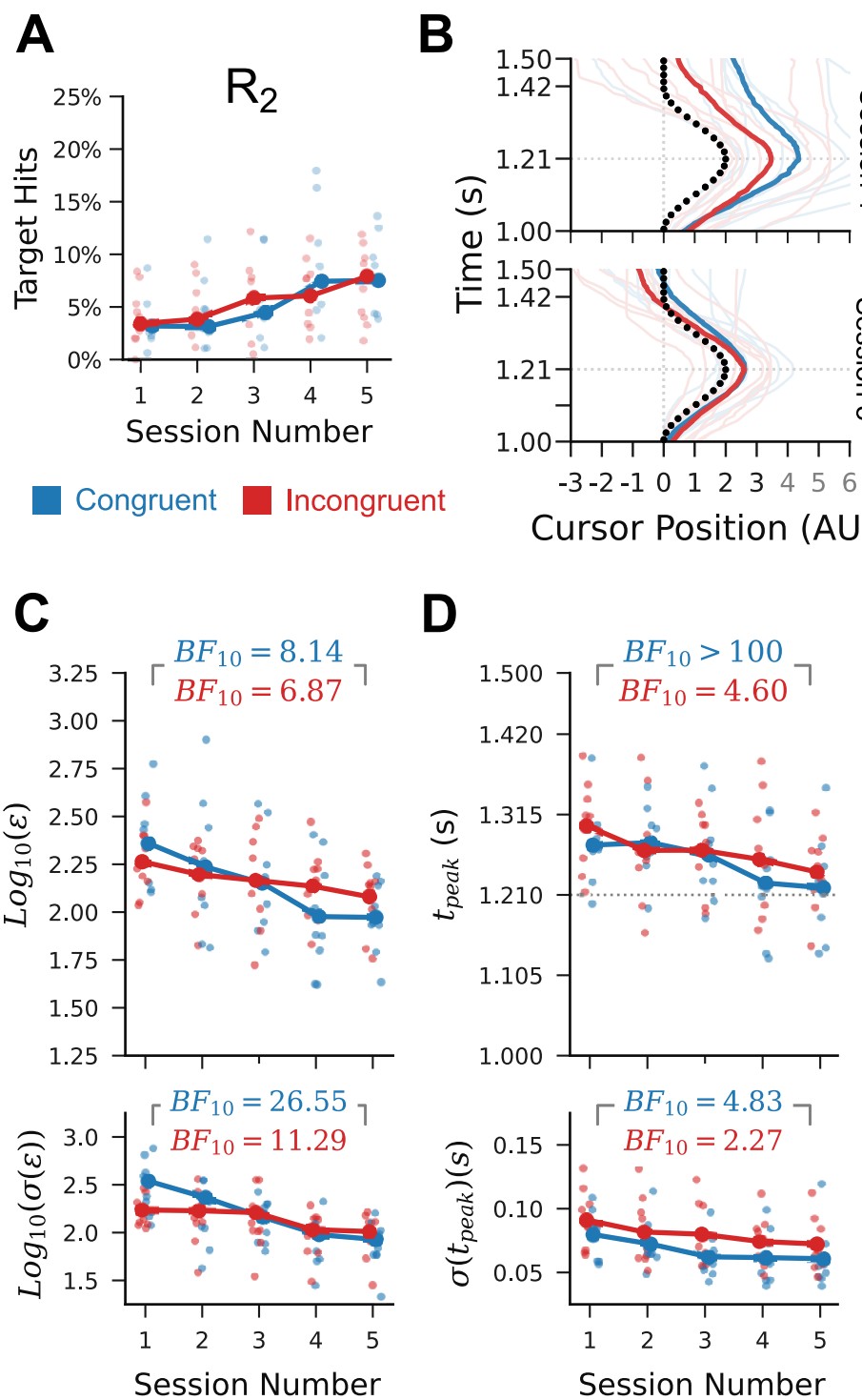

**Fig 4. Performance improvements transferred to the untrained $R_2$ condition.** (A) Peak-aligned target hit percentage for $R_2$. Session 1 to 5 increase is 3.8% (SD = 4.5%) for congruent; 3.3% (SD = 3.5%) for incongruent (B) Changes in median $R_2$ trial trajectory shape for individual participants (faint lines) and across participants (dark lines). (C) Changes in the logarithm of the posterior mean (top) and standard deviation (bottom) of peak-aligned RMS error in cursor trajectory for individual participants (faint points) and marginalising across participants (dark lines). Inset Bayes factors are in favour of a reduction in the marginal values from session 1 to session 5. (D) Similar to C, but for trajectory peak time. All data is from "no-feedback" trials.

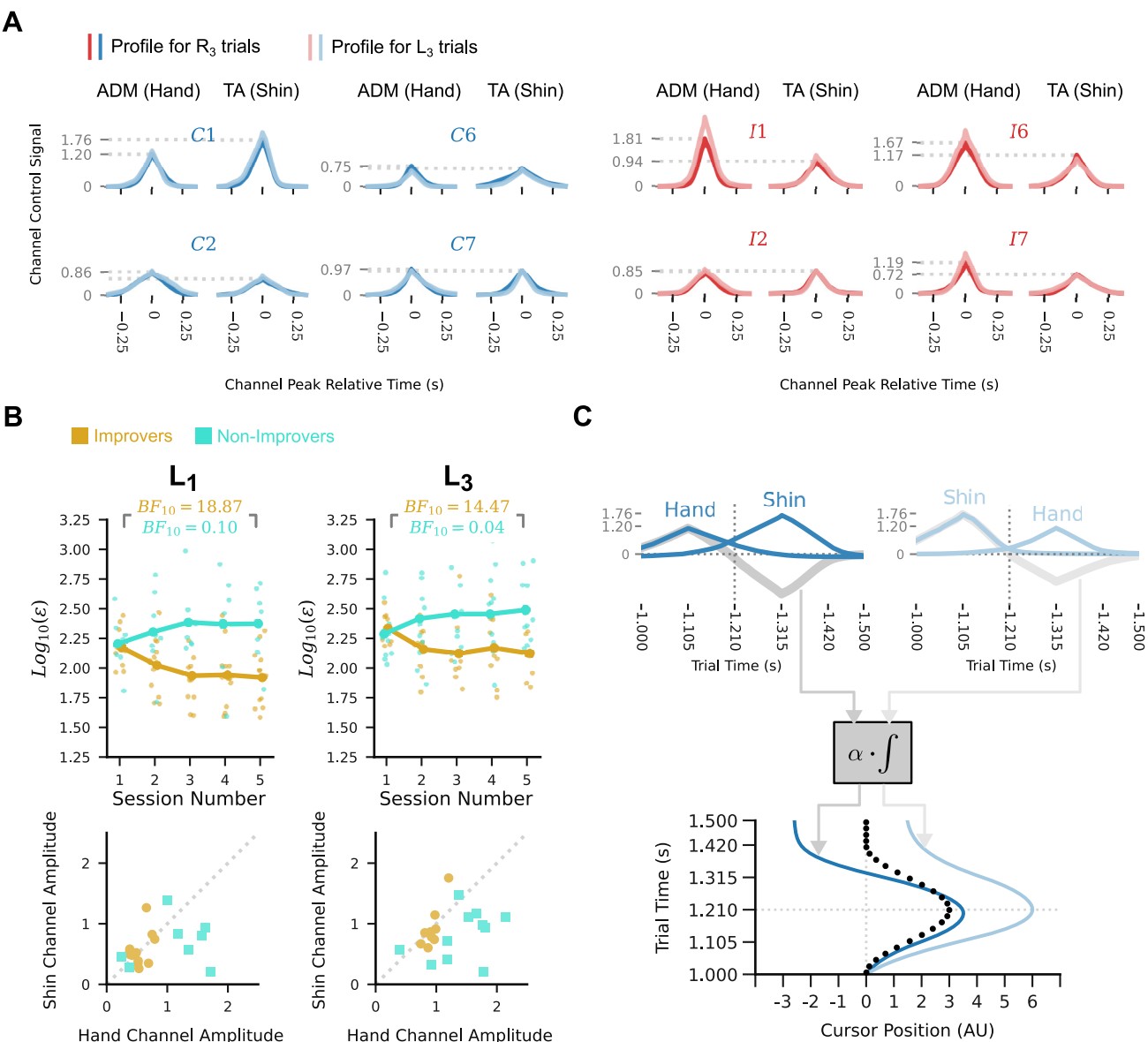

**Fig 5. Reuse of idiosyncratic channel profiles explains the observed patterns of generalisation.** (A) Mean peak-aligned per-participant channel profiles in $R_3$ trials (dark lines) and $L_3$ test trials in session 5, for four example participants. Dotted lines indicate the mean peak amplitudes for the $R_3$ trials. Red traces correspond to incongruent group participants, and blue traces correspond to congruent group participants. Participants produced very similar per-channel outputs for the leftward and rightward path conditions. Results for other sessions and path magnitudes are qualitatively similar. (B) Top row shows trends in cursor trajectory posterior mean RMS error for the small- and large-amplitude leftward paths as in Fig 3, but grouping participants by whether they improved from session 1 to session 5 (yellow circles) or did not improve (turquoise squares). Participants were classified as improving if their Bayes factor in favour of reduction in posterior mean RMS from session 1 to session 5 was greater than 3. Participants who showed an improvement for the untrained conditions tended to have smaller and more similar mean peak amplitude in the hand and shin channel than did the non-improvers. (C) An example illustrating transfer of peak timing and non-transfer of trajectory shape to leftward conditions for a participant with different per-channel amplitudes. The example participant's mean channel profiles for $R_3$ trials (top left) and for $L_3$ trials (top right) are very similar within channel but different across channels. For the rightward path condition, the hand channel is activated first, and the shin channel second, while the order is reversed for the leftward path condition. This leads to two different velocity profiles (top, grey lines), even when the peaks of the channels in each condition occur at the same two times. The cursor velocities are integrated and scaled to give the output cursor trajectory. This results in a different cursor trajectory in each condition, even though they used near identical per-channel outputs and relative timing. The timing of the trajectory peak is almost unchanged in each condition, due to the symmetry of the channel profiles.

Consistent with the preceding explanation, we observed that participants whose trajectory shape improved for the small and large-amplitude leftward conditions between sessions 1 and 5 tended to have smaller and more similar channel peak amplitudes (Fig 5B). Conversely, participants whose trajectory shapes did not improve tended to have larger and less similar channel peak amplitudes. This dissimilarity of channel amplitudes is a sufficient but not necessary condition for a lack of improvement in the leftward cursor trajectory shapes. These observations explain why we observed transfer of improvements in peak timing to the leftward path conditions but did not observe transfer of improvements in cursor trajectory shape to the leftward condition.

## Participants did not learn to generate entirely new outputs

The performance improvements observed in the preceding sections are consistent with improved trajectory shape and reduced variability in the generated shape. We now consider whether this was achieved by developing the ability to generate entirely new outputs that could not be generated at the beginning of training.

To determine if the patterns of hand and shin muscle activity in session 5 could already be generated in session 1, we compared the amplitude and peak timing for R1 and R2 in session 5 to those of all paths in session 1. For each trial in session 5, we computed the minimum difference in amplitude between this trial and all others in session 1. For each participant, we then took the 99th percentile of these trial minima. We reasoned that, if this selected minimum was small, 99% of trials of this scale in session 5 involved channel outputs which could already be generated (to within this small error) in the first session. The same calculation was used for peak timing.

For both amplitude and peak time in both the hand and shin channels, we observe that outputs very similar to those used in session 5 could already be generated during the first session (Fig 6). This suggests that the observed improvements in task performance were not due to the participants learning to generate new per-channel outputs.

We note that this similarity is not limited to the trained paths. S6 and S7 Figs show the same analyses repeated for the $L_1$ and $L_3$ as well as the $R_2$ and $L_2$ conditions, again with high similarity between each session 5 trial and a trial in session 1. Moreover, the similarity is not limited to our choice of features. Using functional principal components analysis (fPCA) we computed a three-dimensional feature space representation of each generated hand and shin profile, and repeated the similarity analysis using these features (S7 Fig). With these features, as with our manually selected ones, the majority of outputs generated in session 5 could already be produced in session 1.

## Participants had different condition-specific biases in channel peak amplitude and timing

An alternative explanation for the limited transfer of trajectory shape improvements to the leftward conditions is that these trajectories may be intrinsically more difficult to generate than the corresponding magnitude rightward trajectories. If so, performance on these conditions during the first session should be worse than that of the rightward conditions. We observed no such bias in the difference in log-RMS for rightward and leftward path trials of equal magnitude (S3A Fig), with individual participants instead showing idiosyncratic biases distributed approximately evenly around zero. This suggests that the limited transfer of trajectory shape quality to the leftward conditions is not a consequence of intrinsic differences in difficulty between the two directions.

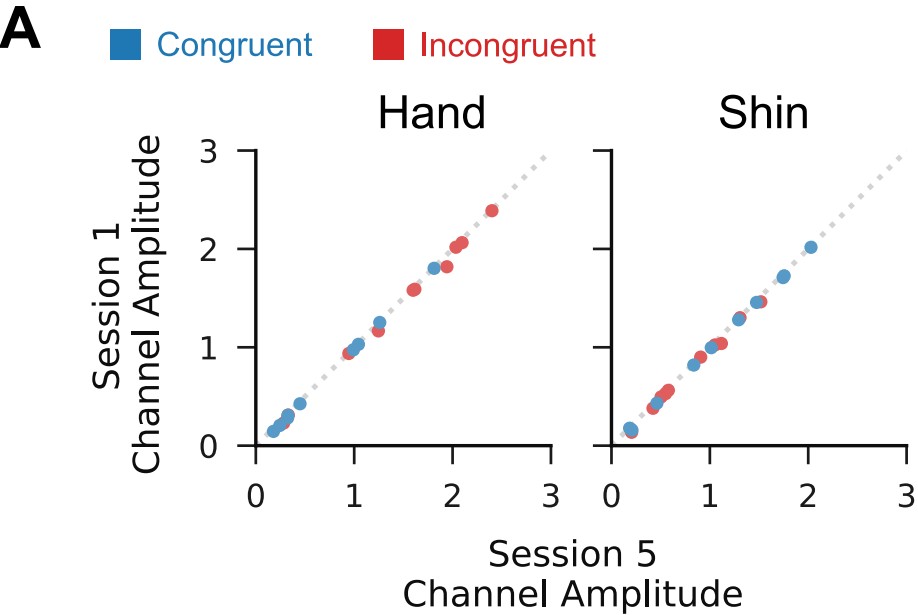

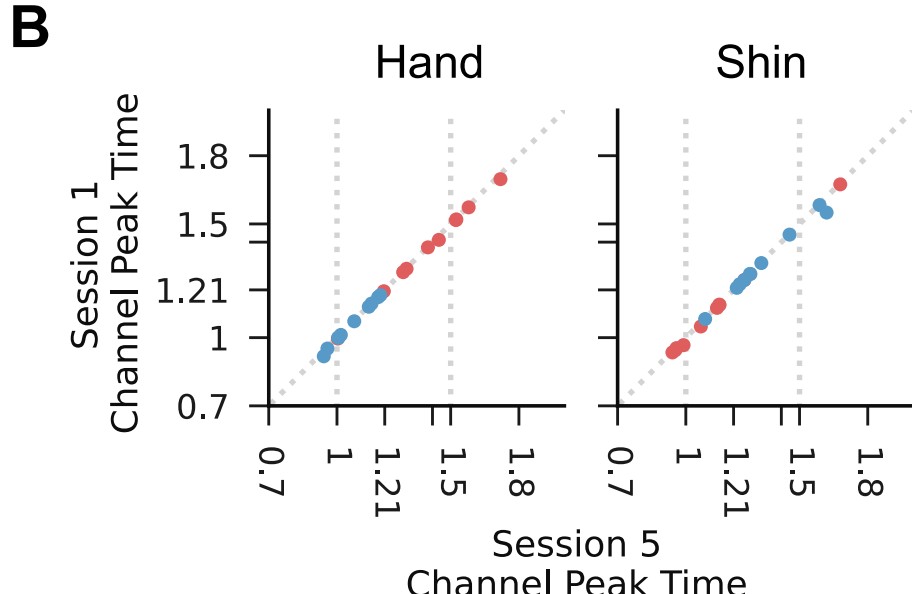

**Fig 6. Participants could produce per-channel outputs in session 1 which closely resembled those used in session 5.** (A) Points show the channel peak amplitude of a selected $R_3$ or $R_1$ trial from session 5 plotted against that of a trial of any condition from session 1. The trials were selected by computing the minimum difference between the channel amplitude of a given trial in session 5 and the channel amplitudes of all trials in session 1, then selecting from these minimums the pair whose magnitude difference was the 99th percentile value. Plotted points therefore represent trials in session 5 for which the minimum difference between the session 5 channel peak amplitude and any session 1 channel peak amplitude is greater than 99% of other similarly calculated differences. 99% of other points (not shown) would therefore have even greater similarity between session 5 and session 1 than the shown points. The consistently small difference between the session 5 and session 1 values demonstrates that participants could produce outputs matching the amplitude of those used in session 5 during the first session. (B) As in A, but for per channel peak time. Again, the peak times in session 5 could be generated during the first session.

A condition-dependent bias was observed for trajectory peak time in session 1. Participants from the congruent group tended to have a later peak time for leftward than for rightward trials, while the pattern was reversed for participants from the incongruent group (S3B Fig). This bias is likely a result of differences in the strength of inputs generated by each input channel. If participants tended to activate the shin channel more vigorously than the hand channel, this would result in an earlier trajectory peak when the shin channel was activated first and later when it was activated second. Consequently, the different congruence conditions will show opposite biases in peak time for the leftward and rightward conditions, due to their opposite mappings from muscle laterality to cursor velocity. Several participants showed such a difference in channel peak amplitudes which persisted in session 5 (S5 Fig).

## Relative timing of the channel profiles changed little with practice

We next assessed whether the timing of the input channel profiles could have contributed to the observed performance improvements. To generate an appropriately timed cursor trajectory, the input channel profiles must themselves be appropriately timed. This could be achieved by triggering the channel inputs relative to some fixed movement initiation time, or by timing the second-activating channel relative to the first. To check for evidence of the latter case, we estimated the correlation between channel peak times after subtraction of movement initiation time. For the trained conditions, the resulting correlations were reliably positive for all participants in both the first and last sessions, with no consistent change in the posterior mean correlation coefficient between these sessions across participants (Fig 7A). Similar positive correlations are seen for the corresponding leftward paths, suggesting that the strategy of relative timing was consistently applied regardless of task condition, and was not strongly affected by training.

Although the second-activating channel is timed relative to the first-activating channel, the interval between activation of the two channels may vary across sessions without affecting the observed correlations. Improvements in trajectory peak time could therefore have been influenced by changes in the interval between channel activations. To check for such a change, we computed the posterior mean difference in channel peak times for each participant in each session and trial condition. The corresponding Bayes factors broadly support no change in cross-participant marginal mean inter-peak interval between sessions 1 and 5 (Fig 7B).

## Discussion

We found that participants gradually improved both the shape and timing of cursor trajectories over five consecutive days of practice. This is notable, as this task had never been practised by the participants before the first session, and we selected muscles which are rarely coordinated together in natural movements. The observed improvements in performance involved minimal changes to the relative timing of the per-channel outputs, instead arising primarily from improvements in the generation of condition-appropriate channel profiles. Notably, the profiles generated in the final session could already be generated during the first session, suggesting that participants did not learn to generate entirely new motor commands.

Distinctive patterns of transfer were observed for conditions practised without post-trial feedback. In particular, while improvements in the timing of the cursor trajectory peak were observed in all path conditions, improvements in cursor trajectory shape were unclear or absent in the leftward path conditions. We explained these observations based re-use of per-channel profiles across both the trained and untrained movement directions (Fig 5C). Based on these observations, we concluded that performance on the path following task was independently influenced by both the timing and the amplitude of the channel outputs, but that

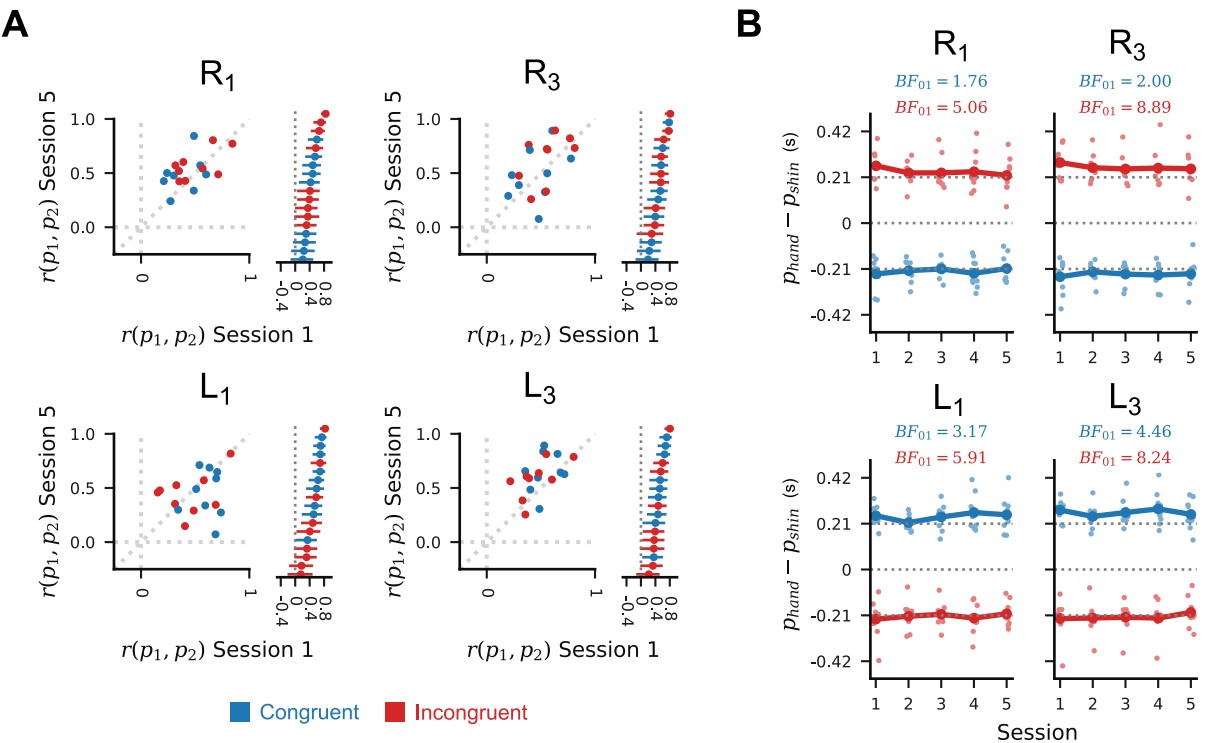

**Fig 7. Channel peaks are timed relative to each other, and inter-peak interval is consistent across sessions.** (A) Scatter plots show the per-participant marginal posterior correlation coefficients between first and second channel peak times in session 1 (horizontal axis) versus those in session 5 (vertical axis). Stacked lines show 99% posterior credible intervals around the session 1 correlation coefficient for each participant. Blue markers are for participants in the congruent group, while red markers are for participants in the incongruent group. (B) Points show posterior mean differences in per-channel peak times (i.e. inter-peak time intervals) for each path scale condition across sessions, in the test blocks. Dotted lines at ±0.21s are ideal inter-peak times. Inset Bayes factors represent evidence in favour of no change in the inter-peak intervals for sessions 1 and 5.

improvements in performance were mainly attributable to changes in the latter. We now discuss what these results imply about how the participants learned new controllers for this task.

## New controllers from old commands

One possible means of learning a new controller is to develop the ability to generate entirely new task-specific motor commands. The process of generating motor commands could be implemented in various ways, including through spatiotemporal muscle synergies [17–19] or dynamical modes in motor-cortical neural populations [20–23]. While details of neural implementation will determine how the controller is learned, a common behavioural signature of learning to generate new motor commands can be found in each case: the participants develop the ability to generate new spatiotemporal patterns of muscle activity. As we recorded EMG signals from single muscles, and used these to directly control the task state, we were able to assess whether new patterns of muscle activity appeared with practice. Across participants, we found no evidence that the muscle activity generated in the final session was different from that which could already be generated during the first session. As such, we conclude that even if the participants did learn to generate entirely new motor commands, such commands were not necessary to achieve the observed performance improvements.

An alternative means of learning a new controller is to explore the space of pre-existing motor commands and select a set of task-appropriate commands to use in different conditions.

Consistent with this explanation, we observed gradual reductions in the standard deviation of both trajectory shape and trajectory peak timing. However, this reduction in variability is also consistent with improvements in the reliability of the selection process. If participants quickly determined which commands to generate for a given task condition, they may still have had to learn to reliably select those commands within the time constraints of the task. The tendency of participants to re-use their idiosyncratic per-channel profiles in the trained and untrained conditions suggests an important role for selection in learning of a new controller. However, it remains unclear from our results whether learning an association between tasks and commands contributed more to developing the new controller than did learning to reliably produce the associated commands.

A related though easily overlooked question for de novo learning is how it balances reuse of existing learning with acquisition of new learning. If new controllers are learned in isolation from existing ones, prior learning cannot be applied to speed up learning of new tasks; when learning to tie our shoelaces and to write by hand, control of the fingers would have to be learned twice. Alternatively, if there is too much overlap between the new and existing controllers, learning one task could lead to changes in performance on the other [24,25]; mastering cursive could help or hinder our ability to tie a bow. In the present study, the tendency of participants to reuse previously learned commands could arise in part from an adaptive bias: preferentially reusing existing learning during learning of new tasks could prevent existing skills from being harmed by modification of their underlying processes. Future studies could more directly assess whether such reuse arises by choice, perhaps owing to the reduced cognitive demand of selecting a well-practiced command, or by necessity, perhaps arising from the basic properties of the neural representation of sensorimotor skill [11,26].

## Independence of selection and timing

Another distinctive feature of our results was independent changes in channel profile shape and relative timing. While profile shapes changed with practice, their relative timing remained largely consistent, including in untrained conditions. Previous studies of sequence learning have suggested that, when the elements of the sequence overlap in time, the later elements are likely to be timed relative to the state of the preceding element, rather than relative to a common movement initiation time [27]. In the context of the path-following task, we observed positive correlations in channel peak times in all task conditions, consistent with relative timing.

As the two channels usually have different profiles (Figs 5A and S5), partly due to the different physiological characteristics of the two muscles, using the precise state of one muscle to trigger activation of the other would generally not produce the same timing when the order of the muscles was reversed. Although we concluded that the second-activating channel is timed relative to the first-activating channel during all five sessions, this does not imply that the second-activating channel is timed relative to the intensity of activity in the first-activating channel. Rather, the second-activating channel may have been timed relative to some qualitative feature of the first-activating channel profile, such as its peak or the time at which power in that channel started to reduce after the peak. It should be noted that this feature-relative timing could be achieved during planning or based on feedback received during execution. Further studies will be required to test these possibilities.

The observed independence of channel peak timing and amplitude is consistent with results from neuroimaging studies of discrete sequence learning. For discrete finger presses, the production of ordered output sequences is attributed to a hierarchical representation in which the complete sequence is built up from successively smaller sub-sequences [28,29]. Separate

representations of the order and timing of finger press sequences are found in bilateral premotor areas during movement preparation, with integrated representations of both order and timing arising in primary motor areas contralateral to the active hand during sequence execution [30–32]. We would anticipate a similar independence in the neural representation of order and timing for the two muscles used in the present study. This suggests a plausible neural basis for independent improvements in the timing and the selection of amplitudes for coordinated motor outputs, though we observed changes in only the latter.

As the second-activating channel is timed relative to the first-activating channel, the peak time of the generated cursor trajectory is influenced by both the time interval between the two channels and the onset time of the first channel. We found little evidence for a change in the average time interval between channel peaks in any condition (Fig 7B). This suggests that the main channel timing-related behavioural feature affecting trajectory peak timing was the timing of the onset of the whole movement. Although the rightward and leftward direction paths require activation of a different muscle at the start of the movement, we observed that improvements in the timing of the cursor trajectory achieved by practising the $R_1$ and $R_3$ conditions were preserved in the untrained leftward conditions. This suggests that the mechanisms responsible for onset timing are at least partly independent of the muscle being activated. This may contribute to explaining why improvements in trajectory peak timing transferred to the untrained leftward conditions despite their requiring reversed order of muscle activation relative to the rightward conditions.

## Factors limiting performance improvement

Although the observed improvements in trajectory shape and timing for the trained conditions are statistically robust, the magnitude of these improvements is often rather small. Despite extensive training over five sessions, participants were unable to reliably hit more than around 15% of the target points for the trained paths. We suggest that this limited effect size is attributable to two main factors.

Firstly, the intrinsic variability of the EMG interface may have limited participants' ability to control cursor movements. This possibility is supported by the observed differences in performance on the small and large-magnitude trained paths. As the magnitude of the rectified EMG signal's envelope is directly related to the variability of the raw EMG signal [33], stronger muscle contractions produce more variable signals than weaker ones. This causes the larger magnitude control signals in our interface to be more variable than smaller magnitude ones, resulting in worse performance on the large-magnitude trained paths compared to the small-magnitude ones. It is plausible that a different type of interface, such as one using limb position or force e.g. [34,35] may support faster or better learning through lower intrinsic variability. While the use of an EMG interface allows convenient access to any muscle of the body–an advantage for studying de novo learning of previously unpractised movements–this advantage should be weighed against the need for precision in the resulting interface.

A second potential reason for the limited learning in our study is the absence of online visual feedback of cursor position during the training trials. As described in Methods, online feedback was restricted to ensure that our study met two important requirements: firstly, that the strategies used by participants to complete training and test trials were consistent; and secondly, that learning during the test trials was limited. Providing online visual feedback during both training and test trials could have resulted in learning during the latter, confounding our analysis of generalisation. Providing online visual feedback during only the training trials could have caused participants to behave differently on these trials compared to the test trials. Importantly, while our decision to limit online visual feedback allowed us to study

generalisation of learning to untrained conditions, it could also have biased participants towards reuse of existing motor commands. In the absence of online visual feedback, learning may result in changes to feedforward controllers more than feedback controllers [36]. As such, it is plausible that the observed tendency to reuse existing motor commands would not arise in tasks with online visual feedback and where feedback controllers were used. Future studies may investigate this possibility by comparing learning and generalisation both with and without online visual feedback.

## Is this learning "De Novo"?

Together, the above-described results suggest that learning of a new continuous control task can be achieved by improving the selection and timing of outputs that are already in the repertoire of the learner. It remains unclear, however, if this is true of de novo learning tasks in general, or if it is a consequence of specific features of the task used in this study. Compared to previous de novo learning studies, our task has several distinguishing features.

By using a combination of muscles which are not typically coordinated in natural movements, we were able to reduce the influence of prior experience on learning. This contrasts with several previous de novo learning paradigms [5,13,14] in which well-practiced movements were deliberately used to reduce the need for exploratory learning of mapping from body state to task state. The learning observed in our study may therefore have a larger component of exploration, but should also be less affected by interference from pre-existing associations. To empirically assess the influence of this type of interference on learning, we assigned participants to either a congruent or incongruent mapping condition. We observed qualitatively similar patterns of learning in both cases, though participants from the incongruent group tended to have more variable performance throughout. This result demonstrates that prior associations between body state and task goals can affect learning, even for previously unpractised tasks.

Another distinctive feature of our task is that the EMG interface controlled the velocity of the output cursor rather than its position. Using the velocity control interface, the path-following task could be completed by generating a pair of appropriately timed pulses of EMG activity. The mechanisms involved in learning to generate a well-timed sequence of discrete motor outputs are likely to differ from those involved in learning more continuous control tasks [37]. As such, although the learning observed in this study meets the definition of de novo learning, the mechanisms supporting learning in the path-following task may not be identical to those observed in other de novo learning studies. It is already well understood that sensorimotor learning is supported by multiple interacting learning processes [38], and we suggest that de novo learning should be similarly understood as arising from a range of learning mechanisms, differently recruited by different tasks.

Our results demonstrate that learning a new controller for an unfamiliar coordinated control task need not involve learning to generate entirely new motor commands. Instead, independent changes in the timing and selection of already available commands may be sufficient to support the production of novel movements.

## Supporting information

**S1 Fig. Demonstration of trajectory peak alignment.** Alignment of the peak of the observed cursor trajectory with the peak of the target trajectory results in more target hits. Throughout the reported analyses, we use the peak-aligned trajectories to compute metrics of trajectory shape quality.
(DOCX)

**S2 Fig. Participants applied untrained condition-specific outputs or re-used existing non-specific outputs to achieve transfer of performance gains to the untrained $R_2$ condition.** A) KDE approximated distributions of observed per-channel peak amplitudes in session 5 for two example participants (C4, congruent; I8 incongruent), separated by trial condition. Vertical lines indicate the modes of each distribution. The distributions for these participants are clearly distinguishable. B) Distributions as in A, but for two other participants (C6 congruent, I2 incongruent) showing extensive overlap for the three trial conditions. C) All participants were assigned to one of two groups based on whether the peak amplitude distributions in at least one of the two channels were distinguishable. Participants were assigned to the "distinguishable" group (N = 7; 3 congruent) if the modal amplitudes for the three conditions were all more than 0.2 apart and assigned to the "indistinguishable" group otherwise (N = 11; 6 congruent). Plots show posterior Log-RMS marginalised across all participants in each distinguishable/indistinguishable group (purple points and lines). Red points are marginal posterior means for participants in the incongruent group, blue for the congruent group. Inset Bayes factors are in favour of a reduction from session 1 to session 5. D) Similar to C, but for trajectory peak time. Participants I3 and C8 are excluded from all analyses in this figure.
(DOCX)

**S3 Fig. Participants had idiosyncratic biases in error across the two trajectory directions, but neither the rightward nor leftward paths were intrinsically more difficult.** (A) Marginal posterior distributions for the difference in log-RMS between the rightward and leftward "no-feedback" trials. Individual horizontal lines are per-participant 95% posterior credible intervals. Shaded curves represent posterior density of the difference across all participants. Red features represent participants from the incongruent group, blue features represent participants from the congruent group. Columns correspond to different trajectory magnitudes. (B) Marginal posterior distributions for the difference in trajectory peak time between the rightward and leftward "no-feedback" trials. Features are as in A.
(DOCX)

**S4 Fig. Per-participant mean channel profiles were re-used across trajectory directions.** Plots show the mean peak-aligned per-participant channel profiles in $R_3$ trials (dark lines) and $L_3$ test trials in session 5. Dotted lines indicate the mean peak amplitudes for the $R_3$ trials. Red traces correspond to incongruent group participants, and blue traces correspond to congruent group participants. Results for other sessions and path magnitudes are qualitatively similar.
(DOCX)

**S5 Fig. Participants could produce per-channel outputs for $L_1$ and $L_3$ in session 1 which closely resembled those used in session 5.** (A) Points show the channel peak amplitude of a selected $L_3$ or $L_1$ trial from session 5 plotted against that of a trial of any condition from session 1. The trials were selected by computing the minimum difference between the channel amplitude of a given trial in session 5 and the channel amplitudes of all trials in session 1, then selecting from these minimums the pair whose magnitude difference was the 99th percentile value. (B) Similar to A, but for per channel peak time.
(DOCX)

**S6 Fig. Participants could produce per-channel outputs for $R_2$ and $L_2$ in session 1 which closely resembled those used in session 5.** (A) Points show the channel peak amplitude of a selected $R_2$ or $L_2$ trial from session 5 plotted against that of a trial of any condition from session 1. The trials were selected by computing the minimum difference between the channel amplitude of a given trial in session 5 and the channel amplitudes of all trials in session 1, then

selecting from these minimums the pair whose magnitude difference was the 99th percentile value. (B) Similar to A, but for per channel peak time.
(DOCX)

**S7 Fig. Participants could produce per-channel outputs for all path shapes in session 1 which closely resembled those used in session 5.** (A) Upper plots show the top three functional principal components of the hand channel profiles, computed using the hand profiles from all no-feedback trials in which the order of hand and shin channel peaks was correct. These three components are sufficient to explain 89.3% of the variance in generated hand channel profiles across all sessions and participants. Each generated channel profile can therefore be well approximated by a linear combination of these three features. We project each participant's hand channel profiles onto each component to get a three-dimensional representation of the hand channel profiles generated on each trial. We then assess the similarity of these components in session 5 and session 1, as explained in S6 and S7 Figs. In this case, we simultaneously compare all six path conditions in session 5 to all six conditions in session 1. As before, we observe that, for 99% of the trials, the generated hand channel profile in session 5 is very similar (in fPCA feature space) to a hand channel profile generated during session 1. (B) Analyses are as in A, but using the shin channel profile. The three selected fPCA components explained 85.8% of the variance in shin channel profiles across all participants and no-feedback trial conditions.
(DOCX)

## Author Contributions

**Conceptualization:** George Gabriel.

**Formal analysis:** George Gabriel.

**Investigation:** George Gabriel.

**Methodology:** George Gabriel, Faisal Mushtaq, J. Ryan Morehead.

**Resources:** Faisal Mushtaq, J. Ryan Morehead.

**Software:** George Gabriel.

**Supervision:** Faisal Mushtaq, J. Ryan Morehead.

**Visualization:** George Gabriel.

**Writing – original draft:** George Gabriel.

**Writing – review & editing:** George Gabriel, Faisal Mushtaq, J. Ryan Morehead.

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
