## [Decision Letter · Decision Letter 0]

15 Jul 2024

Dear Dr Gabriel,

Thank you very much for submitting your manuscript "De Novo Sensorimotor Learning Through Reuse of Movement Components" for consideration at PLOS Computational Biology.

As with all papers reviewed by the journal, your manuscript was reviewed by members of the editorial board and by several independent reviewers. In light of the reviews (below this email), we would like to invite the resubmission of a significantly-revised version that takes into account the reviewers' comments.

We cannot make any decision about publication until we have seen the revised manuscript and your response to the reviewers' comments. Your revised manuscript is also likely to be sent to reviewers for further evaluation.

Sincerely,

Shlomi Haar, PhD

Academic Editor

PLOS Computational Biology

Daniele Marinazzo

Section Editor

PLOS Computational Biology

Reviewer's Responses to Questions

**Comments to the Authors:**

Reviewer #1: This study employs a new task to investigate de novo learning and generalization. The task involves coordinating the activation of two muscles (from different limbs on opposite sides of the body that do not normally co-activate) to control the movement of a cursor along several trajectories, only two of which are trained, and these are on the same side. A third trajectory on the same side and three mirror target trajectories on the opposite side are used to measure generalization. The researchers found that improvements in performance and generalization in this task likely reflect improvements in the selection and timing of muscle activation, suggesting that the "novel" controllers acquired during this learning process are made up of pre-existing controllers. The results are interesting, novel, and contribute significantly to a still largely understudied yet very broad topic of de novo learning.

I have no concerns or constructive suggestions. The only recommendation I would make is to include a link to a video demo of the task, showing the cursor and target trajectories during the training and test trials. It's a bit hard for me to imagine exactly how it works, and since it's novel, I may not be the only one. It can also be added to the osf repository but I think a hyperlink to the video would be best.

Reviewer #2: This papers examines a topic of increasing interest – how we learn new motor skills from scratch, rather than adapting an existing one. The paper is well written, with the introduction providing good overview of the current state of understanding, and motivation for the current experiments.

The results are quite clear. The actual extent of learning is a bit weak, even over five days of training. The data are analyzed very carefully, however, and the authors do a very thorough job of determining in detail exactly how participants are behaving in their experiment. In particular, they establish that participants improve their performance largely by modulating the timing of stereotyped muscle activations, and show how this accounts for the preserved timing to untrained leftward trajectories, but poorer overall performance. Overall, I think this is a very sound contribution to our understanding of how people approach learning a challenging, novel motor skill.

Major comments:

A lot of interesting and important material appears only as supplementary materials. Personally I did not find that it was possible to obtain a clear understanding of the findings without consulting these supplementary figures. So I would suggest that a lot or at least some of this be incorporated into the main figures in the paper.

The choice to have participants perform the task without online visual feedback is slightly surprising. It's not clear why this was done, and I expect that it would significantly impact participants' ability to learn to control the cursor and likely prompt them to learn it in a different way compared to if online feedback were shown. This requires a bit more justification and discussion of the implications of this choice on what conclusions can be drawn and how these findings relate to other studies of de novo learning. This is, for me, perhaps the most salient distinctive feature of this particular task, along with the extremely limited set of training trajectories. These limitations aren't mentioned in the discussion but should be. It would be interesting, in the future, to see how learning might differ with realtime feedback and given a richer variety of target trajectories to train on.

Minor:

Telgen, Parvin, Diedrichsen, ( J Neurosci 2014) ought to be cited somewhere as conceptually a very important paper that first highlighted the distinction between adaptation and de novo learning.

In the Introduction: "Secondly, the participant may learn a new association between task goals and existing motor commands (Golub et al., 2018) ... Thirdly, the participant may have to learn to reliably and rapidly select task-appropriate commands from their existing repertoire. " It's not clear to me exactly what the difference is between these two ideas. 'Task-appropriate' seems to mean more or less the same thing as being contingent on 'task goals', and 'commands from an existing repertoire' = 'existing motor commands'. And learning an 'association' versus 'learning to rapidly select' just don't seem very distinct ideas. Please clarify these points.

Reviewer #3: The paper investigates how new sensorimotor skills, specifically those involving continuous and coordinated control of relatively unpracticed muscle movements, are learned. Using a myoelectric EMG interface, participants learned to control a computer cursor with two muscles over five consecutive days. Improvements in task performance, such as timing and trajectory shape, were observed with post-trial visual feedback. The study suggests that the new controllers acquired during this learning process are constructed from existing motor commands rather than entirely new ones. This innovative approach provides insights into "de novo" learning, highlighting the potential for constructing new motor skills from existing components.

Overall, the study is well designed and has the potential contribute to the body of knowledge related to motor control in the context of skill motor learning. There are, however, a few issues that need to be addressed in order for this manuscript to be considered for publication.

Major Concerns.

(1) The use of Bayes factor instead of more common statistical methods like ANOVA (assuming normal distribution of the data) is a major weakness. The authors should justify their choice and possibly include traditional statistical analyses to strengthen their findings.

(2) The protocol included three train-test mini-sessions per day over five days, with five test trials per session. This might have led to learning during the test trials, confounding the generalization results. A more careful analysis is required to address this potential issue.

(3) I believe that a control experiment to test the re-use hypothesis is missing. To verify if participants indeed produced the optimal pattern needed to succeed in the task already from Day 1, a new group should be tested but learn different pathways (e.g., R2 and L2) and then be examined for generalization and improvement to test the re-use hypothesis.

(4) Figure 4- The correlation between peak time and amplitude might be achieved through various means (since a point from each measure was selected). Precise comparison of the entire path as done with RMS error is needed. This needs further clarification and analysis.

Minor Concerns

(1) The phrase "Such skills cannot be learned by adjusting an existing control policy" in the Significant Statement is an overstatement. It should be tempered and rewritten for accuracy.

(2) Add the experimental protocol to Figure 1 for clarity.

(3) Page 18, line 367: The text should indicate "reduced error" instead of "increased accuracy".

(4) Page 33, line 435: The phrase "...whose trajectory shapes did not improve..." is inaccurate as some participants did improve. Figure 3B also needs clarification regarding the classification of participants who did not improve but aligned with the correlation line.

PLOS authors have the option to publish the peer review history of their article (what does this mean?). If published, this will include your full peer review and any attached files.

Reviewer #1: **Yes: **Denise

Reviewer #2: No

Reviewer #3: No

**Have the authors made all data and (if applicable) computational code underlying the findings in their manuscript fully available?**

Reviewer #1: **No: **They promise to do so but haven't yet

Reviewer #3: **No: **
---

## [Decision Letter · Decision Letter 1]

16 Sep 2024

Dear Dr Gabriel,

We are pleased to inform you that your manuscript 'De Novo Sensorimotor Learning Through Reuse of Movement Components' has been provisionally accepted for publication in PLOS Computational Biology.

Best regards,

Shlomi Haar, PhD

Academic Editor

PLOS Computational Biology

Daniele Marinazzo

Section Editor

PLOS Computational Biology

Reviewer's Responses to Questions

**Comments to the Authors:**

Reviewer #1: I am satisfied with the revisions.

Reviewer #2: The authors have provided thoughtful and thorough responses and revisions. I have no further concerns.

typo on line 259 - the word "from" appears twice

Reviewer #3: The authors have adequately addressed all of my comments, and I am satisfied with the revisions made. I believe the paper is now ready for publication, and I have no further comments. Thank you for your efforts.

**Have the authors made all data and (if applicable) computational code underlying the findings in their manuscript fully available?**

Reviewer #1: Yes

Reviewer #2: Yes

Reviewer #3: Yes

PLOS authors have the option to publish the peer review history of their article (what does this mean?). If published, this will include your full peer review and any attached files.

Reviewer #1: **Yes: **Denise Henriques

Reviewer #2: No

Reviewer #3: No

---

## [Editor Report · Acceptance letter]

27 Sep 2024

PCOMPBIOL-D-24-00867R1 

De Novo Sensorimotor Learning Through Reuse of Movement Components

Dear Dr Gabriel,

I am pleased to inform you that your manuscript has been formally accepted for publication in PLOS Computational Biology. Your manuscript is now with our production department and you will be notified of the publication date in due course.

With kind regards,

Olena Szabo
